# MATHEMATICAL REASONING IN LATENT SPACE

**Dennis Lee, Christian Szegedy, Markus N. Rabe, Kshitij Bansal, and Sarah M. Loos**
Google Research
Mountain View, CA, USA
`{ldennis,szegedy,mrabe,kbk,smloos}@google.com`

## ABSTRACT

We design and conduct a simple experiment to study whether neural networks can perform several steps of approximate reasoning in a fixed dimensional latent space. The set of rewrites (i.e. transformations) that can be successfully performed on a statement represents essential semantic features of the statement. We can compress this information by embedding the formula in a vector space, such that the vector associated with a statement can be used to predict whether a statement can be rewritten by other theorems. Predicting the embedding of a formula generated by some rewrite rule is naturally viewed as approximate reasoning in the latent space. In order to measure the effectiveness of this reasoning, we perform sequences of rewrite steps both in formula space and in latent space, and compare the quality of embeddings of the resulting formulas to their predicted embeddings. Our experiments show that graph neural networks can make non-trivial predictions about the rewrite-success of statements, even when they propagate predicted latent representations for several steps. Since our corpus of mathematical formulas includes a wide variety of mathematical disciplines, this experiment is a strong indicator for the feasibility of deduction in latent space in general.

## 1 INTRODUCTION

Automated reasoning has long been considered to require development of logics and "hard" algorithms, such as backtracking search. Recently, approaches that employ deep learning have also been applied, but these have focused on predicting the next step of a proof, which is again executed with a hard algorithm (Loos et al., 2017; Gauthier et al., 2017; Lederman et al., 2018; Bansal et al., 2019b).

We raise the question of whether hard algorithms could be omitted from this process and mathematical reasoning performed entirely in the latent space. To this end, we investigate whether we can predict useful latent representations of the mathematical formulas that result from proof steps. Ideally, we could rely entirely on predicted latent representations to sketch out proofs and only go back to the concrete mathematical formulas to check if our intuitive reasoning was correct. This would allow for more flexible and robust system designs for automated reasoning.

In this work, we present a first experiment indicating directly that theorem proving in the latent space might be possible. We build on HOList, an environment and benchmark for automated theorem provers based on deep learning (Bansal et al., 2019b), which makes use of the interactive theorem prover HOL Light (Harrison, 1996), an interactive proof assistant. The HOList theorem database comprises over 19 thousand theorems and lemmas from a variety of mathematical domains, including topology, multivariate calculus, real and complex analysis, geometric algebra, and measure theory. Concrete examples include basic properties of real and complex numbers such as $(x^{-1} = y) \Leftrightarrow (x = y^{-1})$, and also well-known theorems, such as Pythagoras' theorem, the fundamental theorem of calculus, Skolem's theorem, Abel's theorem for complex power series, and that the eigenvalues of a complex matrix are the roots of its characteristic polynomial.

We focus on *rewrite rules* (or *rewrites* in short). Rewrites are only one of several proof tactics in HOL Light, but they enable powerful transformations on mathematical formulas, as they can be given arbitrary theorems as parameters. For example, the formula $3^2 = z$ can be rewritten to $3 \cdot 3 = z$ by performing a rewrite with the parameter $x^2 = x \cdot x$. Alternatively, a rewrite may diverge (as it operates recursively) or it may return the same formula – in both these cases we consider the rewrite

Figure 1: Overview of latent space prediction propagated over multiple steps

to *fail*. For instance, in the example above, the rewrite would fail if we used equation $x + y = y + x$ as a rewrite parameter instead, since the expression $3^2 = z$ does not contain any $+$ operators to match with.

In our experiments, we first train a neural network to map mathematical formulas into a latent space of fixed dimension. This network is trained by predicting – based on the latent representation being trained – whether a given rewrite is going to succeed (i.e. returns with a new formula). For successful rewrites we also predict the latent representation of the resulting formula. To evaluate the feasibility of reasoning in latent space over two steps, we first predict the latent representation of the result of a rewrite, then we evaluate whether the predicted latent representation still allows for accurate predictions of the rewrite success of the resulting formula. For multi-step reasoning beyond two steps, we predict the future latent representations based on the previous latent representation only - without seeing the intermediate formula (cf. Figure 1). Our experiments suggest that even after nine steps of reasoning purely in latent space, neural networks show non-trivial reasoning capabilities, despite not being trained on this task directly.

## 2 RELATED WORK

Our work is motivated by deep learning based automated theorem proving, but is also closely connected to model based reinforcement learning and approaches that learn to predict the future as part of reinforcement learning.

Model based reinforcement learning is concerned with creating a model of the environment while maximizing the expected reward (e.g. Ha & Schmidhuber (2018)). Early works have already shown that predicting the latent representations of reinforcement learning environments with deep learning is sometimes feasible - even over many steps (Oh et al., 2015; Chiappa et al., 2017; Burda et al., 2019). This can enable faster training, since it can preempt the need for performing expensive simulations of the environment. Predicting latent representation was also proposed in Brunner et al. (2018) as a regularization method for reinforcement learning.

One recent successful example of model based reinforcement learning is Kaiser et al. (2019), where the system learns to predict the pixel-wise output of the Atari machine. However this approach is based on actually simulating the environment directly in the "pixel space" as opposed to performing predictions in a low dimensional semantic embedding space. More related to our work is Piotrowski et al. (2019), which attempts to learn to rewrite simple formulas. The goal is there again is to predict the actual outcome of the rewrite rather than a latent representation of it. In Dosovitskiy & Koltun (2017), they predict "expected measurements" as an extra supervision in addition to the reward signal.

Graph neural networks have been used for premise selection in higher order logic (Wang et al., 2017) and more recently by Paliwal et al. (2020) as the core component of DeepHOL, a neural theorem prover for higher order logic. In this work, we build upon their neural network architecture, but utilize it for a different task.

## 3 HOL LIGHT

HOL Light (Harrison, 1996) is an *interactive proof assistant* (or interactive theorem prover) for higher-order logic reasoning. Traditionally, proof assistants have been used by human users for creating formalized proofs of mathematical statements manually. While proof assistants have some limited forms of automation, it is still a cumbersome process to formalize proofs, even when they

are already available in natural language. Some large scale formalization efforts were conducted successfully in HOL Light, Coq (Coq) and Isabelle/HOL and other theorem provers, for example the formal proof of the Kepler conjecture (Hales et al., 2017), the four color theorem (Gonthier, 2008), and the correctness of a microkernel (Klein et al., 2009). They required significant meticulous manual work and expert knowledge of interactive proof assistants.

Lately, there have been several attempts to improve the automation of the proof assistants significantly by so called "hammers" (Kaliszyk & Urban, 2015). Still, traditional proof automation lacks the mathematical intuition of human mathematicians who can perform complicated intuitive arguments. The quest for modelling and automating fuzzy, "human style" reasoning is one of the main motivations for this work.

## 3.1 Rewrite Tactic in HOL Light

The HOL Light system allows the user to specify a *goal* to prove, and then offers a number of *tactics* to apply to the goal. A tactic application consumes the goal and returns a list of subgoals. Proving all of the subgoals is equivalent to proving the goal itself. Accordingly, if a tactic application returns the empty list of subgoals, the parent goal is proven.

In this work, we focus on the rewrite tactic (`REWRITE_TAC`) of HOL Light, which is a particularly common and versatile tactic. It takes a list of theorems as parameters (though in this work we only consider applications of rewrite with a single parameter). Parameters must be an equation or a conjunction of equations, possibly guarded by a condition. Given a goal statement $T$ and parameter $P_i$, the rewrite tactic searches for subexpressions in $T$ that match the left side of one of the equations in $P_i$ and replaces it with the right side of the equation. The matching process takes care of variable names and types, such that minor differences can be bridged. The rewrite procedure is recursive, and hence tries to rewrite the formula until no opportunities for rewrites are left. The rewrite tactic also has a set of built-in rewrite rules, representing trivial simplifications, such as $\mathrm{FST}(x, y) = x$. Note that `REWRITE_TAC` uses "big step semantics", meaning that the application of each individual operation can perform multiple elementary rewrite steps recursively. For more details on `REWRITE_TAC`, refer to the manual (HOL Light Rewrite Tactic Reference).

## 4 Reasoning in Latent Space

In this section we describe our training data, a model architecture, and an evaluation setup that allows us to demonstrate mathematical reasoning in latent space. We use the embedding model by Paliwal et al. (2020) to map formulas into a fixed dimensional latent space, and train the model on two prediction tasks: (1) on the outcome (success or failure) of a large number of possible formula rewrite operations, and (2) on the embedding of the outcome of the rewrite. For evaluation, we use the embedding prediction not only for predicting the embedding of the next formula, but also the formulas that result from several successive rewriting steps.

The challenge in the design of the model is preventing the embeddings from collapsing. Trained naïvely to minimize the $l^2$ distance between the predicted and the true embedding of the result of the rewrite, the embedding model quickly learns to map most formulas to the same embedding - destroying the information needed for performing multiple rewrite steps in latent space. To address this problem, we stop the gradient from updating the embedding of the outcome formula directly (but we share the embedding network for the source expression).

We have also found that adding extra noise to the embedding during training makes the formulas more robust to prediction errors for multi-step step reasoning. We hypothesize that training with noisy embeddings acts as a fuzzy error-correction mechanism by preventing errors from accumulating as quickly during multiple steps of reasoning in the latent space. Both ideas are described in detail in Subsection 4.3.

## 4.1 Training Data

We generate the training data from the theorem database of the HOList environment (Bansal et al., 2019b), which contains 19591 theorems and definitions ordered such that later theorems use only earlier statements in their human proof. The theorems are split into 11655 training, 3668 validation,

and 3620 testing theorems. To generate our training data, we generate all pairs $(T, P)$ of theorems from the training set, where $P$ must occur before $T$ in the database (to avoid circular dependencies). We then interpret theorem $T$ as a goal to prove and try to rewrite $T$ with $P$ using the `REWRITE_TAC` of HOL Light. This can result in three different outcomes:

1. $T$ is rewritten by theorem $P$ successfully, and the result differs from $T$.

2. The rewrite operation terminates, but fails to change the input theorem $T$.

3. The rewrite operation times out or diverges (becomes too big).

In our experiments, we consider only the first outcome as successful, i.e. when the application finishes within the specified time limit and changes the target, as a successful rewrite attempt. Each training example, therefore, consists of the pair $(T, P)$, the success/fail-bit of the rewrite (1 for successful rewrites, 0 for failed rewrites), and, for successful tactic applications, the new formula that results from the rewrite, which we denote with $R(T, P)$. Of the 115 million possible pairs of $(T, P)$ where $T$ falls in the training split, there are 1.6 million successful rewrites that are used as training examples.

## 4.2 MODEL ARCHITECTURE AND TRAINING METHODOLOGY

Let $S$ denote the set of syntactically correct higher-order logic formulas in HOL Light. Additionally, we define two latent spaces, $L = \mathbb{R}^k$ and $M = \mathbb{R}^k$ ($k = 1024$), corresponding to embeddings of theorems used as goals, and the statements used as parameters, respectively.

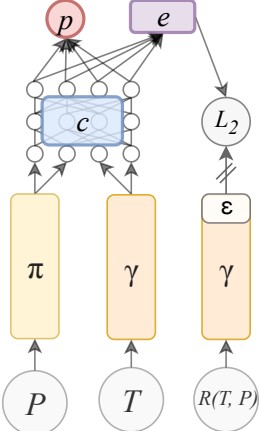

We use two embedding networks $\gamma : S \longrightarrow L$ and $\pi : S \longrightarrow M$, one for each of the two formulas $T$ and $P$. For both embedding networks, we follow the GNN approach by Paliwal et al. (2020): We interpret the formulas' abstract syntax trees as graphs, feed the graphs into graph neural networks, and let the networks exchange messages between neighboring nodes for a fixed number of hops. The final embeddings of the nodes are max-pooled to form a representation of the entire formula in a fixed dimensional embedding space $\mathbb{R}^k$.

Figure 2: Depiction of the network architecture

The embeddings are then concatenated and processed by a three-layer perceptron $c : L \times M \longrightarrow C$, which is followed by $p : C \longrightarrow \mathbb{R}$, a single output linear function producing a logit predicting whether theorem $P$ rewrites $T$. The perceptron $p$ is trained with logistic regression on the success/fail-bit of the rewrite. Formally, this output is $p(c(\gamma(T), \pi(P)))$, which we also refer to as $p(T, P)$ or $p(\gamma(T), \pi(P))$.

The output of $c$ is also used as input to another linear function $e : C \longrightarrow L$, which predicts the latent representation of the result $\gamma(R(T, P))$ by minimizing the $l^2$ distance. Formally, this output is $e(c(\gamma(T), \pi(P)))$, but we also use $e(T, P)$ or $e(\gamma(T), \pi(P))$.

## 4.3 STOP GRADIENTS AND NOISE

Note that we stop gradient backpropagation between the $l^2$ loss and the second embedding tower $\gamma(R(T, P))$. The purpose of this is to prevent the second tower from making the magnitude of the embedding of the rewrite outcome very small or "collapse". We discuss alternative solutions to this problem in Section 5.

In the case of the second goal embedding tower $\gamma$, we add Gaussian noise $\varepsilon$ with 0 mean and standard deviation $\beta = 10^{-3}$ during training only. We can see in the experimental section that training with noise has the effect of more accurate downstream predictions in multiple step reasoning.

## 4.4 REASONING

After we have trained our model on the training set theorems (and theorem pairs) of the HOList benchmark, we can use it to perform rewrites in the latent space alone.

We use $p : S \times S \longrightarrow \mathbb{R}$ as a quality metric for the propagated embedding vector. Given an approximation $E_T'$ of the latent representation $\gamma(T)$, we can evaluate $p(E_T', \pi(P))$ for a large number of tactic parameters $P$. This is compared with true rewrite successes of $T$ by $P$ to assess the quality of the approximation.

To evaluate multiple steps of reasoning in the latent space, start with the embedding of formula $T_1$ and predict the embeddings that result from rewriting by theorems $P_1, \cdots P_k$ in that order. To assess the quality of the overall reasoning, the same sequence of rewrites is performed with the actual formulas and the final predicted embedding is evaluated with respect to the formula resulting from the sequence of formal rewrites.

In latent space $L = \mathbb{R}^k$ we start from some initial theorem $T_1 \in S$. The following schema indicates the sequence of predictions performed in latent space $L$:

$$T_1 \xrightarrow{\gamma} l_1 \in L \xrightarrow{e(\bullet, \pi(P_1))} l_2 \in L \xrightarrow{e(\bullet, \pi(P_2))} l_3 \in L \xrightarrow{e(\bullet, \pi(P_3))} \cdots$$

This is compared with the following formal sequence of rewrites:

$$T_1 \xrightarrow{R(\bullet, P_1)} T_2 \xrightarrow{R(\bullet, P_1)} T_3 \xrightarrow{R(\bullet, P_3)} \cdots$$

That is, $l_2$ approximates the latent vector of $T_2$, that is $\gamma(R(T_1, P_1))$ and $l_3$ approximates the latent vector of $T_3$, that is $\gamma(R(R(T_1, P_1), P_2))$, by construction.

## 5 ALTERNATIVE MODEL ARCHITECTURES

Learning a high quality embedding space while simultaneously learning to make predictions in that space proved to be some challenge. As we described above, the embeddings quickly approach 0 if trained naïvely, minimizing the least squares distance between the $e(T, P)$ and $\gamma(R(T, P))$ without improving the quality of $p$. Through the course of this work we evaluated several alternative approaches to tackling this problem. The empirical comparison between these architectures is shown in Section 6.4.

### 5.1 FIXED LATENT SPACE

Our first, straightforward approach to this problem was to fix the embedding space $L$ and learn predictions from a separate space $L'$. The advantage of this approach is that it does not suffer from embedding collapse during the prediction of the embedding of the rewrite outcome. On the other hand, this approach incurs training of three different models, which is awkward and cumbersome. The three models are shown in Figure 3.

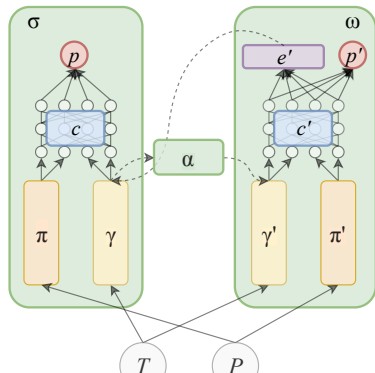

Figure 3: Depiction of the network architecture

1. Rewrite success prediction $\sigma : S \times S \longrightarrow \mathbb{R}$,
2. Rewrite outcome prediction $\omega : S \times S \longrightarrow \mathbb{R} \times L$,
3. Embedding alignment prediction $\alpha : L \longrightarrow L'$.

The rewrite success prediction model $\sigma(T, P)$ is identical to the function $p$ described above. Training $\sigma$ fixes the embedding space $L$, from which we predict rewrite success.

In addition to $\sigma$, we train a separate rewrite outcome prediction $\omega(T, P)$. This model has an identical two-tower embedding architecture as $\sigma$, but with an extra prediction layer $e'$ to predict the embedding vector of the outcome of the rewritten formulas in $L$. Here the embedding towers are denoted by $\gamma'$ and $\pi'$, the combiner network is $c'$ and the two linear prediction layers are $p'$ and $e'$. That is: $\omega(T, P) = (p'(c'(\gamma'(T), \pi'(P))), e'(c'(\gamma'(T), \pi'(P))))$. We denote the embedding space learned by $\omega$, $L'$.

Since $\sigma$ and $\omega$ produce latent vectors $\gamma(T) \in L$ and $\gamma'(T) \in L'$ in different spaces, we need to align those spaces to enable deduction purely in the embedding space. In particular, $\gamma(R(R(T, P_0), P_1))$

is approximated by $e(\gamma'(R(T, P_0)), \pi'(P))$, and so we need $\gamma'(R(T, P_0))$. For that, we train a translation model $\alpha : L \longrightarrow L'$, which is a three layer fully connected network with 1024 units in each layer and rectified linear activations, which predicts $\gamma'(T)$ given an embedding of $\gamma(T)$. Note that $\alpha$ does not see $T \in S$ as an input, it makes its prediction based on the latent space representation $\gamma(T)$ of $T$ alone.

Reasoning with this model can be summarized by:

$$T_1 \xrightarrow{\gamma'} l'_1 \in L' \xrightarrow{e'(\bullet, P_1)} l_2 \in L \xrightarrow{\alpha} l'_2 \in L' \xrightarrow{e'(\bullet, P_2)} l_3 \in L \xrightarrow{\alpha} l'_3 \in L' \xrightarrow{e'(\bullet, P_3)} \cdots$$

## 5.2 CENTERING AND NORMALIZATION

A different approach to learning a good embedding prediction is normalizing the embeddings to a fixed length. The naïve implementation of this leads both $\gamma$ and $e$ to predict an almost-constant embedding vector independent of $T$, with the relevant information encoded in the noise. Centering the embeddings before normalization mitigates this issue. In our experiments, we found that using a variation of batch normalization without learned mean and scale parameters, followed by $l_2$ normalization, is sufficient to force the model to predict meaningful embeddings from $e$. However, this model is sensitive to hyperparameters, in particular the scaling of losses between $p$ and $e$, and even in the best configuration we found performance was worse than our final model.

## 5.3 STOCHASTIC EMBEDDINGS

Another natural approach is to replace the mean squared error objective of the embedding prediction output $e$ with a KL divergence objective, which is scale invariant. To do this, we attempted to substitute the deterministic embedding tower $\gamma$ with a stochastic variant, where the final layer of the tower is used to parameterize a diagonal Gaussian distribution. The final layer of the embedding network $e$ is similarly used to paramaterize a diagonal Gaussian. $KL(e(T, P) || \gamma(R(T, P)))$ is the objective to minimize.

This model proved difficult to optimize: the naïve solution fails to converge. A carefully tuned information bottleneck mitigates this problem, but the performance of the model remains poor.

# 6 EXPERIMENTS

This section provides an experimental evaluation that aims to answer the following question: Is our model capable of predicting useful embedding vectors multiple steps ahead? We explore the prediction quality of the embedding vectors (for rewrite success) and see how the quality of predicted embedding vectors degrades after $e$ is used for predicting multiple steps in the latent space alone.

## 6.1 NEURAL NETWORK ARCHITECTURE DETAILS

Our network has two independent towers, which are each 16-hop graph neural networks with internal node representations in $\mathbb{R}^{128}$. The output of each of the two towers is fed into a layer that expands the dimension of the node representation to $\mathbb{R}^{1024}$ with a fully connected layer with shared weights for each node. This is followed by maximum pooling over all nodes of the network. We add Gaussian noise with $\beta = 10^{-3}$ to the embedding of the goal theorem. The two resulting embedding vectors are concatenated along with their element-wise multiplication, and are processed by a three layer perceptron with 1024 hidden units and rectified linear activation between the layers.

Our model is trained with $g = 16$ choices of $T$ in each batch. For each $T_i$, there are $l + 1 = 16$ choices of $P$: one successful example in each group and $l$ random negatives. However all choices of $P$ corresponding to other $T_{k \neq i}$ are used as negative instances as well, regardless of whether they would rewrite it – this is justified by the fact that only a few theorems rewrite any given $T_i$ so this introduces only a small amount of uncorrelated label noise. This training methodology is motivated by the fact that evaluating the combiner network is much cheaper than computing the embedding using the graph neural network. Based on Alemi et al. (2016), we expect that hard negative mining, or iteratively replacing random negative examples with examples that $p(T, P)$ classifies incorrectly, would improve our results significantly, but it is left for future work.

## 6.2 EVALUATION DATASET

In order to measure the performance of our models after multiple deduction steps are performed, we generate datasets $D_0, D_1, \ldots, D_r$ successively by applying rewrites to a randomly selected set of examples. We start with all theorems from the validation set of HOList, denoted by $\mathbf{T_0}$. For each theorem $T_{i,j} \in \mathbf{T_i}$, we perform all possible rewrites $R(T_{i,j}, P_k), k < j$ and include them in $D_i$. We then randomly select a new goal to include in $\mathbf{T_{i+1}}$. This method ensures that the goals in $\mathbf{T_i}$ accurately represent the full range of complexity of theorems present in the database, while the distribution of parameters $P$ and the ratio of successful rewrites by each parameter matches that of the space of all possible rewrites.

## 6.3 EVALUATION OF REWRITE PREDICTION MODEL

To evaluate a model $m : S \times S \longrightarrow \mathbb{R}$ on dataset $D_i$, we compute $m(T, P)$ for each pair $(T, P)$ in the dataset. We expect pairs where $R(T, P)$ succeeds to score highly relative to pairs where $R(T, P)$ fails. For a concrete metric, we compute the ROC curve of the predictions and use the area under the curve (AUC) as our main metric. Higher curves and higher AUC values represent more accurate predictions of rewrite success. We measure how ROC curves change as we use different approximations of $\gamma(T)$.

As we have trained our model only on pairs of theorems from the original database, the model exhibits increasing generalization error as we evaluate it on formulas with increasing number of rewrites. The accuracy of the the model $p$ evaluated directly on theorems after multiple rewrites is referred to as the "True" measurement. This gives an upper bound on the rewrite success prediction since noisier embedding vectors generated by $e$ will produce worse results on average.

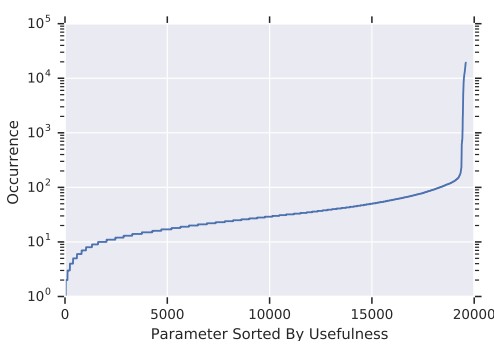

Additionally, we want to measure the rate at which the latent vectors degrade as we propagate them in embedding space as described in Subsection 4.4. Embedding predictions generated directly from theorem statements, $e(\gamma(T), \pi(P))$, are referred to as "One Step", while predictions produced through propagating are referred to as "Multi Step". A comparison of these generation methods is shown in Figure 4.

Figure 5: The distribution of successful rewrites for each parameter $P$, computed over all $(T, P)$ pairs in the database. Note that a small portion of parameters can be used for many theorems. This allows the Random and Usage baselines to achieve above-chance performance by predicting success based on parameter alone.

In order to evaluate the performance of our model in isolation we need to compare it with carefully selected baselines:

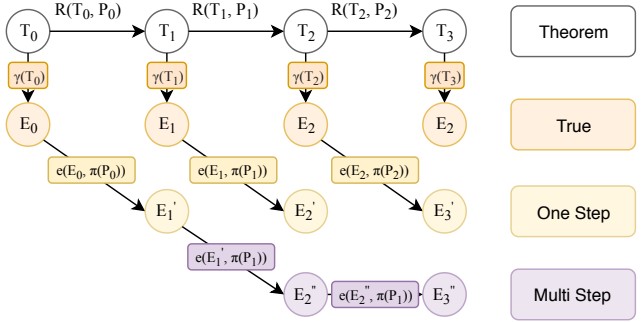

Figure 4: Detailed view of embedding generation scheme for the True, Pred (One Step), and Pred (Multi Step) curves.

1. As Figure 5 shows, a few theorems are much more frequently applicable than others. We want to see how the prediction of rewrite success performs based on the rewrite parameter alone if we ignore the theorem to be rewritten. One way to establish such a baseline is to feed a randomly selected theorem $T'$ to $p$ instead of $T$ to predict its rewrite success. This is referred to as the "Random" baseline.

2. A stronger "baseline" is achieved by utilizing the ground-truth rewrite statistics to rank each parameter $P$ by the number of theorems it successfully rewrites. This is the best achievable prediction that does not depend on $T$. We refer to this as the "Usage" baseline.

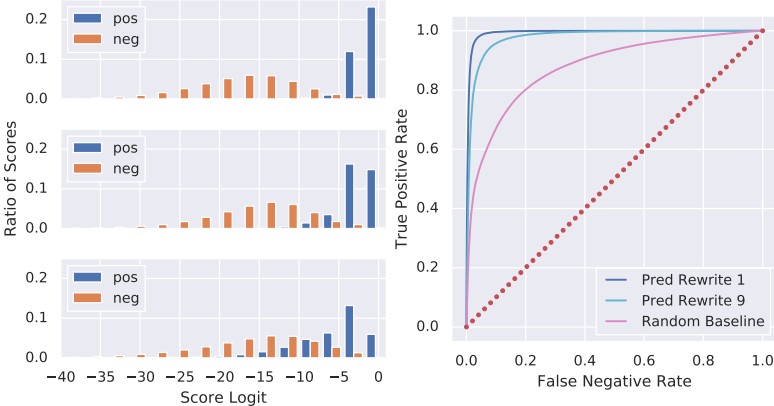

Figure 6: Left: Histograms of scores for first step of rewrite predictions (top), ninth step of rewrite predictions (middle), and random baseline (bottom). Right: Corresponding ROC curves.

Figure 6 shows the distribution of the theorem pair prediction score logits of $p(c(\gamma(T), \pi(P)))$, for the those "positive" pairs that rewrite and the "negative" pairs that do not rewrite. Note that the ratio is normalized separately for the positive and negative pairs as negative pairs occur much more frequently than positive pairs.

One can see that the quality of the rewrite success prediction degrades somewhat after nine steps of reasoning purely in the latent space, but it is still much better than the random baseline. This gives clear evidence that the embedding prediction manages to propagate much useful information over multiple steps in the latent space alone.

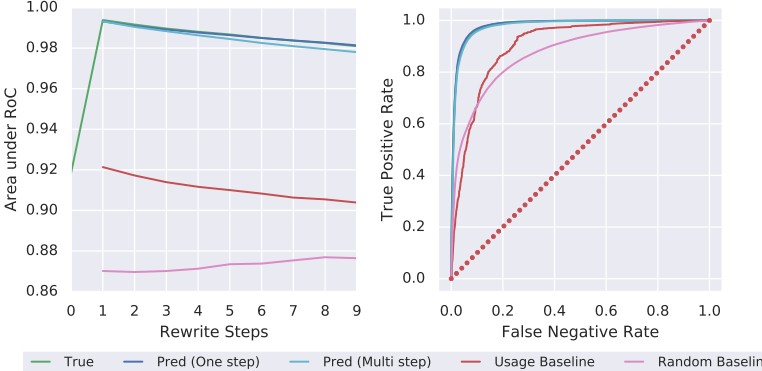

Figure 7: Left: Comparison of Area under ROC curves for each embedding method. Right: Comparison of ROC curves for each embedding method after the ninth rewrite.

In Figure 7 we make further measurements and comparisons on the quality of reasoning in the embedding space. One can see that our model could perform reasoning for nine steps in the embedding space and still retain a lot of the predictive power of the original model.

In order to appreciate the above results one should keep in mind that our model is not trained on statements that were already rewritten. All the training was done only on the theorems present in the initial database. The reduction of prediction performance is apparent from downward trajectory of the "Pred (Multi step)" curve, which isolates this effect from that of the error accumulated by the embedding predictions, the effect of which is measured indirectly by the "True" curve.

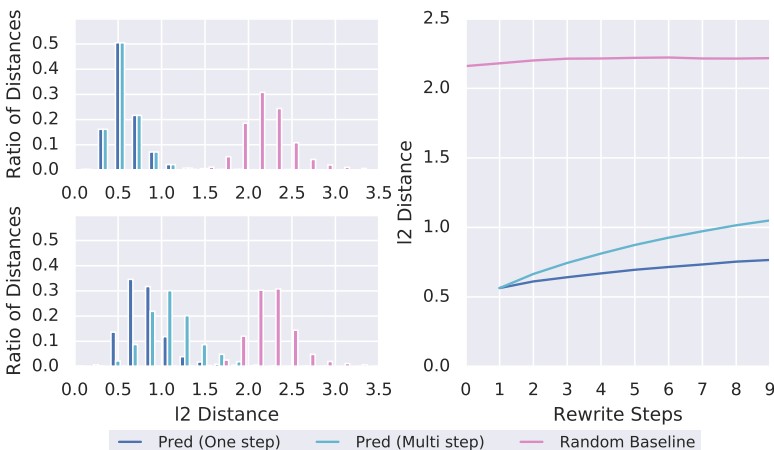

Figure 8: Left: Histogram of $l^2$ distances between each embedding method and the true embedding for rewrite steps 1 (top) and 9 (bottom). Right: Plot showing the mean $l^2$ distances across rewrite steps.

In Figure 8 we measure the $l^2$ distance from the predicted embedding vectors to the true embedding vectors $\gamma(T)$ of formulas after multiple rewrite steps in the latent space. These results are consistent with our earlier findings on success of rewrite predication after rewrite steps in the latent space: while there is some divergence of the predicted embedding vectors from the true embedding vectors (as computed from the rewritten statements directly), the predicted embedding vectors are significantly closer to the true embedding vectors than randomly selected embeddings.

## 6.4 COMPARISON OF ALTERNATIVES

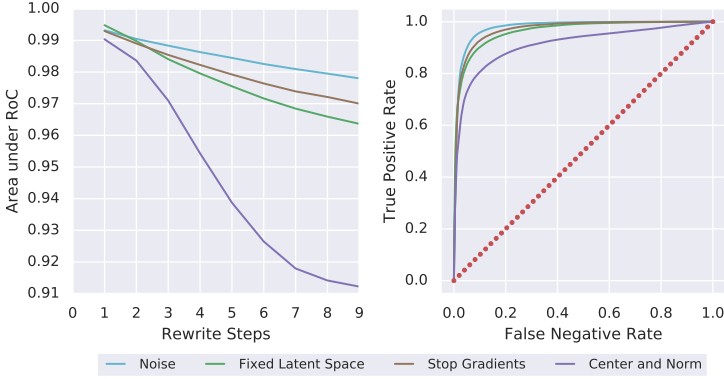

Figure 9: Left: Comparison of Area under RoC curves for each architecture, using predictions fully in the latent space (Pred (Multi Step)). Right: Corresponding ROC curves after the ninth rewrite step.

Our final model adds noise to the embeddings produced by $\gamma$, during training time only. This forces the network to produce embeddings sufficiently far apart to distinguish between them. We found that this method provided the best experimental results compared to the models discussed in Section 5 as shown in Figure 9. We hypothesize that the noise makes the network robust to future prediction mistakes as the prediction of the next rewrite is trained on noisy versions of the theorem embedding. This may act as a natural error correction mechanism. Figure 9 supports this hypothesis, since the difference between the noisy and noiseless versions of the model increases with the number of rewrite steps performed in the latent space. However, our experiments indicate that adding noise alone without stop gradients or other approaches is not sufficient to learn a diverse latent space.

## 6.5 EVALUATION OF REWRITEABILITY

Particularly interesting rewrites are those that change the rewritability of a theorem, or rewrites where $R(T, P)$ fails and $R(R(T, P_0), P)$ succeeds, or vice versa.

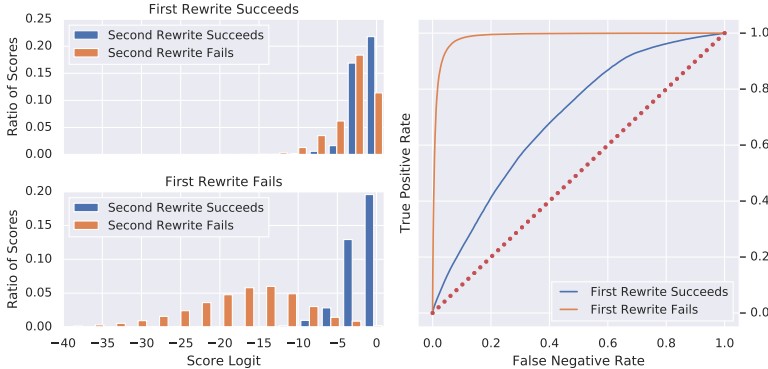

Figure 10: Left: Histogram of scores for $R(R(T, P_0), P)$ where $R(T, P)$ succeeds (top) and where $R(T, P)$ fails (bottom). Right: Corresponding ROC curves.

In Figure 10, we evaluate our model on these cases. We created a database of statements that are the result of $R(R(T, P_0), P)$ and split it into two groups, depending on whether $R(T, P)$ succeeds or fails. Within each group, we score each rewrite and plot the resulting ROC curves. From this figure, we can see that the model is capable of detecting when the rewriteability of a theorem does not change (First Rewrite Succeeds and Second Rewrite Succeeds, or First Rewrite Fails and Second Rewrite Fails).

Most interestingly, cases where the first rewrite fails and the second rewrite succeeds are also scored correctly. This demonstrates that the predicted embeddings are precise enough to encode the appearance of new possibilities. However, it is noteworthy that the model often fails to detect when rewrites are no longer possible. This may be due to label noise caused by randomly sampling negative examples, as some "negative" examples are actually positives, leading to generalization error. A deeper investigation of these failures is an interesting subject for further work.

## 7    CONCLUSION

In this paper we studied the feasibility of performing complex reasoning for mathematical formulas in a fixed $(1024)$ dimensional embedding space. We proposed a new evaluation metric that measures the preservation semantic information under multiple reasoning steps in the embedding space. Although our models were not trained for performing rewrites on rewritten statements, nor were they trained for being able to deduce multiple steps in the embedding space, our approximate rewrite prediction model $p$ has demonstrated significant prediction power on as many as nine rewrite steps in latent space alone. Our methods showcase a simple but general methodology for reasoning in latent space.

This work can also be seen as an easy to use, fast to train, and crisp evaluation methodology for representing mathematical statements by neural networks. We hope that this will lead to improved neural network architectures for mathematics.

It is likely that such representations prove helpful for faster learning to prove without imitating human proofs like that in DeepHOL-Zero Bansal et al. (2019a), given that premise selection is a closely related task to predicting the rewrite success of statements. Self-supervised pre-training or even co-training such models with premise selection could prove useful as a way of learning more semantic feature representations of mathematical formulas.

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

## A    VISUALIZATION OF LATENT SPACES

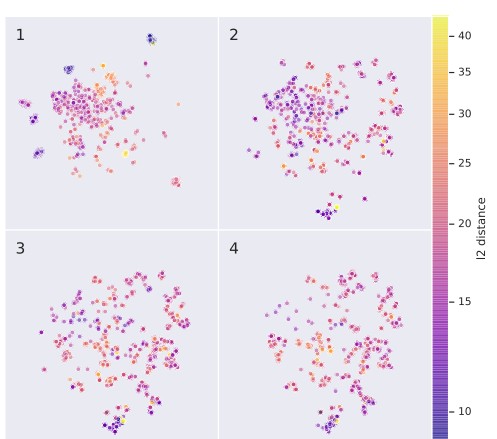 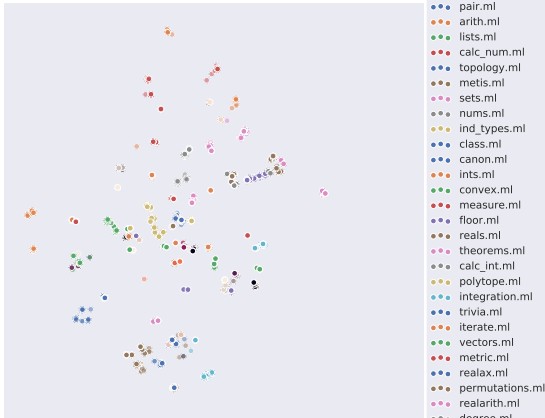

(a) Visualization of the embedding spaces produced by embedding prediction $e$ in the latent space after $1, 2, 3$ and $4$ rewrite steps. The points are colored by their $l^2$-distance to the true embedding $\gamma(T)$.

(b) Visualization of the embedding space $\gamma(T)$ evolving over four rewrite steps. Theorems are colored by the area of mathematics they originate from, sourced from the theorem database. The brightness of the embeddings corresponds to the rewrite step in which it was generated, with more recent embeddings being darker.

Figure 11: tSNE visualization of embedding spaces.

In addition to numerically evaluating the performance of our latent embedding models, it is interesting to visually examine the embedding spaces produced. Figure 11a depicts the predicted embeddings of the datasets $D_1 \cdots D_4$ by using PCA to reduce the dimension of each point in the joint dataset to 50, then using tSNE to produce a 2D visualization which is filtered by original step. The points are colored by their $l^2$-distance to the true embedding. From this graph it can be seen that the distribution of embeddings shifts meaningfully over multiple rewrites, further demonstrating that the rewrite task produces non-trivial changes.

We also wished to compare the embeddings produced by our model to human understanding of mathematics. In the HOList database, each theorem comes with tags describing the area of mathematics it originates from, such as topology or set theory. In Figure 11b we separate the embeddings of theorems from these categories by color, and examine the their evolution across multiple rewrites. It can be seen that theorems originating from similar human categories correspond to similar embeddings in the latent space.

