# OpenReview forum: "Mathematical Reasoning in Latent Space"
_ICLR.cc/2020/Conference — Accept (Talk)_

### Official Review · AnonReviewer1 · 2019-10-21
**Official Blind Review #1**

**Rating:** 8

**Review:**

= Summary
Embeddings of mathematical theorems and rewrite rules are presented. An in-depth analysis of the resulting embeddings is presented, showing that a network can learn to "apply" embedded rewrite rules to embedded theorems, yielding results that are similar to the embedding of the rewritten theorem. [i.e., app'(emb(thm), emb(rule)) is near to emb(app(thm, rule))] This is an interesting property for the application of deep learning to automated theorem proving, though not directly a breakthrough result.

= Strong/Weak Points
+ Simply a cute result, showing that proof search can remain in embedding space for a limited time horizon without having to switch back into the theorem prover environment.
+ Nicely designed experiments testing this (somewhat surprising) property empirically
- Missed opportunity of better analysis of which theorem/rewrite rule properties are more likely to fail
- Writing sometimes a bit overcomplicated (e.g., Sect. 4.5 could just be a figure of a commuting diagram and two sentences...)
- Architecture choice unclear: Why are $\sigma$ and $\omega$ separate networks. This is discussed on p4, but it's unclear to me how keeping $\sigma$ separate is benefitial for the analysis, and this is not picked up again explicitly again?

= Recommendation
Overall, this is a nice, somewhat surprising result. The writing and experiments could use some improvement, but I believe that the majority of the ICLR audience would enjoy seeing this result (even though it would have no impact on most people's research)

= Detailed Comments
- page 4, Sect. 4.4: Architecture of $\alpha$ would be nice (more than a linear layer?)
- page 5, paragraph 3: "we from some" -> "we start from some"
- p6par1: "much cheaper then computing" -> than
- p6par6: "on formulas that with" -> no that
- p6par7: "measure how rate" -> "measure the rate"
- p8par1: "approximate embedding $\alpha(e(\gamma'(...)))$ - $e$ is undefined and should probably be $e'$ (this is also the case in the caption of Fig. 5), and $c'$ should probably be included as well. However, I don't understand the use of $\alpha$ here. If Fig. 4 is following Fig. 3 in considering $p(c(\gamma(T), \pi(P)))$, then Fig. 4 should plot the performance of, e.g., $p(c(e'(c'(\gamma'(T_{i-1}), \pi'(P_{i-1}))), \pi(P_i)))$ (i.e., $p$ applied to approximate embedding of $T_i$ and ("true") embedding of $P_i$). I believe that's what "Pred (One Step)" expresses, but it would maybe be generally helpful to be more precise about the notation in Sect. 6.

**Experience Assessment:**

I have published one or two papers in this area.

**Review Assessment: Checking Correctness Of Derivations And Theory:**

N/A

**Review Assessment: Checking Correctness Of Experiments:**

I assessed the sensibility of the experiments.

**Review Assessment: Thoroughness In Paper Reading:**

I read the paper thoroughly.

---

> ### Author Response · Authors · 2019-11-13
> **Thank you for your feedback!**
>
> We thank the reviewer for the great feedback. We have simplified the architecture described in the paper by combining the networks $\sigma$ and $\omega$, and included the results from this architecture as well, producing a more robust architecture that performs better for multiple rewrite steps (while keeping the original, more complicated solution as one of the baselines).
>
> As suggested, we have added further analysis of failure cases. We also corrected the typos and clarified the definitions of True, Pred (One Step) and Pred (Multi Step) variants.
>
> We are very grateful for the review that helped to improve the paper significantly.

---

> > ### Comment · AnonReviewer1 · 2019-11-13
> > **Discussion of Review#1**
> >
> > Thank you for these changes, which are greatly improving the readability of the paper. I especially appreciate Fig. 5, which makes the experiment design much easier to grasp, and the new analysis in Sect. 6.5. The fact that the new, simpler model architecture works slightly better than the original two-network design is a nice new development as well, and I'm grateful that you included this (substantial, but important) change in the revision. With these improvements, I'm happy to raise my score.

---

### Official Review · AnonReviewer3 · 2019-10-22
**Official Blind Review #3**

**Rating:** 8

**Review:**

The paper proposes a method to do math reasoning purely using formula embeddings. The proposed method employs a graph neural network to embed math formulas to a latent space. The formula embeddings are then combined with theorem embeddings (also formulas, computed in the same way as formula embeddings) to predict whether one can do one step of math reasoning using the corresponding theorem, and also to predict the embeddings of the resulting formula. Empirically the authors demonstrate that the method can be chained end-to-end to do multiple steps of reasoning purely in the latent space.

I tend to accept this paper, (but also OK if it gets rejected), for the following reasons: (1) the idea is novel and interesting; (2) the writing of the paper is below conference standard and very hard to read, especially the method and the experiment sections.

===========================================================================

Novelty and significance

I really like the idea of doing math reasoning in latent space. The idea is definitely novel and interesting. It is related to existing works such as neural logic induction[1] and planning in latent space[2]. It is amazing that one can do multiple steps of math reasoning after only training the model using data from one single step. It would be interesting to see how it can improve existing learning-based theorem provers.

My question is if we want to integrate the proposed method into theorem provers, after multiple steps of math reasoning, how would us know the goal has been proved? Is it possible that we can train a decoder that maps back from the latent space to the formula space? Also can it work with theorems that decompose the current goal into several sub-goals? I know these are not the concerns of this paper, but I would be really grateful if you could provide some intuitive answers!

===========================================================================

Writing

The paper is not well-organized and not written in a consistent way. For the method and the experiment sections, I need to jump back and forth several times in order to understand what the authors are trying to say.

1. Typo: Third paragraph in section 1, "...which is makes use of ...".
2. It's very confusing when the authors introduce \sigma and \omega in the beginning of section 4: why would you need two networks predict the same thing?
3. Mentioning "merging \sigma and \omega, is left for future work" is confusing before formally introducing \sigma and \omega.
4. Even when the authors formally introduce \sigma and \omega in 4.2, it is still not clear that why both of them are used for modelling the success probability.
5. In fact, I don't know why \omega needs to output p. It's never mentioned in the experiment section.
6. The rationale of the two tower design (why not combine two) is not clearly explained.
7. Typo: Page 5 last paragraph, "... negative instances for for each ...".
8. The itemized part in 5.3, "...carefully selected baselines: 1.xxx, 2.xxx, 3. xxx, 4. xxx". However, both 3 and 4 are not baselines!
9. It is not clear that baseline 1 and 2 correspond to which baselines in later experiments.
10. Reading the baselines before the experiments is very confusing. For example, for baseline 1, it is very hard to understand why would we want to use such an unusual baseline, and why it is called a "random baseline".
11. Baseline 2 is actually referred to as "usage baseline" but this name is not introduced in the itemized part.



[1] Rocktäschel, Tim, and Sebastian Riedel. "End-to-end differentiable proving." Advances in Neural Information Processing Systems. 2017.
[2] Srinivas, Aravind, et al. "Universal planning networks." arXiv preprint arXiv:1804.00645 (2018).




**Experience Assessment:**

I have read many papers in this area.

**Review Assessment: Checking Correctness Of Derivations And Theory:**

N/A

**Review Assessment: Checking Correctness Of Experiments:**

I carefully checked the experiments.

**Review Assessment: Thoroughness In Paper Reading:**

I read the paper thoroughly.

---

> ### Author Response · Authors · 2019-11-13
> **Thank you for your feedback!**
>
> We are thankful for the valuable feedback. The main concern of this review is the quality of the writing and experimental details. We updated the paper to clarify all the names and ensure that all terms are introduced before they are used.
>
> The two tower design was necessary since the decision whether theorem T can be rewritten using parameters P requires both pieces of information, so we need to feed them to the network. In fact $\omega$ does not need to predict p, but it gives extra supervision signal and therefore regularizes the prediction. The random baseline is necessary because of the unbalanced nature of the rewrite success, this is hard to control, so we added an extra baseline that shows that our results are better than just ignoring any of the input expressions (theorem or parameter).
>
> In addition, we have simplified the architecture described in the paper by combining the networks $\sigma$ and $\omega$, and included the results from this architecture.
>
> We have significantly improved the experimental section by further clarifying the experiments and expanding them with more supporting measurements. We have also moved the two non-baselines out of the baselines section.
>
> Finally, thank you for the insightful questions! With our current setup, the goal is to simply perform reasoning steps in latent space without specifically proving any statements. There are several approaches to make the network predict a closed goal, for example by predicting a fixed embedding such as the zero vector.
>
> We expect that most semantic aspects of the formula could be recovered, but not superficial features as the naming of the variables should not affect the rewriteability of formulas. The question how much of the formula can be recovered is probably dependent on the theorem database, since only those aspects that manifest in different rewrite successes are expected to be recovered.
>
> We don't have much intuition on the decomposability of embeddings, but it seems like a fascinating research direction.
>
> We are grateful for the feedback which has helped to make the paper much clearer and more readable.

---

> > ### Comment · AnonReviewer3 · 2019-11-15
> > **Reviewer #3 Response**
> >
> > Thank you for the response and the paper revision. Now it looks more readable! I have raised my score accordingly.

---

### Official Review · AnonReviewer2 · 2019-10-23
**Official Blind Review #2**

**Rating:** 8

**Review:**

The paper proposes a technique to perform reasoning on mathematical formulas in a latent space. The model is trained to predict whether a rewrite rule can be applied to a formula given its latent representation. When the rewrite is possible, the model also predicts the embedding of the resulting formula. Experiments show that the network can be applied multiple steps in a row, while operating only in the embedding space.

1. As mentioned in the paragraph before Section 4.1, it would be much simpler to consider a single latent embedding space L. In that case, \sigma and \alpha become unnecessary and we only need to train \omega. Did you try to have a single network? This seems a much more natural approach to me, and I'm surprised that you did not start with that. From my experience, aligning embedding spaces is something that usually does not work very well, especially in high dimension. The role of \sigma seems very redundant given \omega.

2. If you consider \sigma, why do you also predict the rewrite success with \omega? Couldn't it be simply a function from S x S -> L ?

3. The graph neural networks used in the model are not described in the paper, only a reference to Paliwal et al (2019) is given. It would be helpful to have a brief paragraph describing this architecture, for readers not familiar with the referenced paper.

4. How large is the training set of (T, P) pairs? I don't think this is mentioned in the paper.

5. To train \sigma and \omega, the negative instances are selected randomly. You mention that negative mining should improve over this strategy. What does negative mining correspond to in this context? Are there bad rewrites better than others?

6. Did you consider using an inverse function (say G), that maps an embedding in L / L' back to S (i.e. the inverse function of gamma / gamma'). I would imagine that even if an embedding X is a bit noisy, because not exactly equal to gamma(P) where P is the expression it represents, you could consider doing the propagation with gamma(G(X)). This could be a possibility to remove the noise you have when doing multi-step operations (and potentially go way beyond 4 steps). Also, G could be used to check whether you obtain the expected formula after 4 steps, which would be a more informative information than the L2 distance between the resulting embedding and the embedding of the final formula.

Overall, the model is a bit complicated (e.g. question 1.), but the results are promising, the paper is well written, and the ability to manipulate formula embeddings is probably going to be useful in the context of theorem proving.

**Experience Assessment:**

I have read many papers in this area.

**Review Assessment: Checking Correctness Of Derivations And Theory:**

I assessed the sensibility of the derivations and theory.

**Review Assessment: Checking Correctness Of Experiments:**

I assessed the sensibility of the experiments.

**Review Assessment: Thoroughness In Paper Reading:**

I read the paper at least twice and used my best judgement in assessing the paper.

---

> ### Author Response · Authors · 2019-11-13
> **Thank you for your feedback!**
>
> We thank the reviewer for the constructive feedback. The use of a fixed embedding space $L$ and a separate space $L^\prime$ was useful as it naturally prevents the collapse of embeddings. However this could be counteracted by stopping the gradient at the right place in the simplified architecture  which was suggested in the original paper and is now described in the updated paper.
>
> As suggested, we have added further analysis of failure cases, and describe strategies for negative mining from these examples. In addition, we have included a brief description of the graph neural network architecture used in Paliwal et al (2019). We also include further details on the construction of training set.
>
> Training a decoder to predict the results of rewrites from the latent space is an interesting idea, but is technically challenging and we felt it was out of scope for this paper. We managed to counteract the noisiness of predicted embedding by training on noisy embeddings which trains the network to be robust to random changes and improves the prediction of multi-step rewrites significantly.
>
> We are grateful for the suggestions that contributed significantly to improving the quality of the paper.

---

> > ### Comment · AnonReviewer2 · 2019-11-15
> > **Response to rebuttal**
> >
> > Thank you for the response. The updated version of the paper clarified the questions I had.

---

### Author Response · Authors · 2019-11-13
**Updates to the paper**

Addressing the reviewers' suggestions and concerns, we added further experimental results and made significant clarifications to the paper, while striving to keep the broad ideas and findings intact. In summary, the following changes have been made:

1 Introduction:
Added overview of latent space prediction (Fig 1)

4.1 Training Data:
Added statistics of training data.

4.2 Model Architecture and Training Methodology:
Rewritten with a simpler, more effective new model architecture. Previous architecture moved to (5 Alternative Model Architectures). Updated corresponding plots.
Added short description of GNN encoder.

5 Alternative Model Architectures:
New section containing the previous architecture and explaining why we used it.
Description of other possible configurations.
Experimental comparison with alternatives.

6 Experiments
Added figure explaining generation of different curves (Fig 5)
Updated all experiment curves to 9 steps of rewrites (Fig 6, 7, 8)

6.1 Neural Network Architecture Details:
Clarified batching procedure for training data.
Added description of hard negative mining.

6.2 Evaluation Dataset:
Explain improved sampling method (for evaluation data only) which better preserves complex statements.

6.3 Evaluation of Rewrite Prediction Model:
More detailed explanation of experiments.
Moved baselines that were not baselines out of itemized section

6.4 Comparison of Alternatives:
Added experiments comparing the alternative model architectures described in section 5.

6.5 Evaluation of Rewriteability:
Added experiment showing the model’s performance in key cases and a common source of failures.

6.6 Visualization of Latent Spaces:
Added description of plots and analysis

---

### Decision · Program_Chairs · 2019-12-19

**Decision:**

Accept (Talk)

**Comment:**

This paper was very well received by the reviewers with solid Accept ratings across the board.
The subject matter is quite interesting -  mathematical reasoning in latent space, and it was suggested by a reviewer that this could be a good candidate for an oral. The AC agrees and recommends acceptance as an oral. Some of the intuitions of what is being done in this paper could be better visualized and presented and I encourage the authors to think carefully about how to present this work if an oral presentation is granted by the PCs.